# Controlled Release and Cell Viability of Ketoconazole Incorporated in PEG 4000 Derivatives

**DOI:** 10.3390/polym15112513

**Published:** 2023-05-30

**Authors:** Carolina R. Inácio, Gabriel S. Nascimento, Ana Paula M. Barboza, Bernardo R. A. Neves, Ângela Leão Andrade, Gabriel M. Teixeira, Lucas R. D. Sousa, Paula M. de A. Vieira, Kátia M. Novack, Viviane M. R. dos Santos

**Affiliations:** 1Department of Chemistry, Federal University of Ouro Preto, Ouro Preto 35400-000, MG, Brazil; carolina.inacio@aluno.ufop.edu.br (C.R.I.); gabriel.sn@aluno.ufop.edu.br (G.S.N.);; 2Department of Physics, Federal University of Ouro Preto, Ouro Preto 35400-000, MG, Brazil; ana.barboza@ufop.edu.br; 3Department of Physics, Federal University of Minas Gerais, Belo Horizonte 31270-901, MG, Brazil; 4Center for Research in Biological Sciences, Laboratory of Morphopathology, Federal University of Ouro Preto, Ouro Preto 35400-000, MG, Brazil

**Keywords:** ketoconazole, polymers, controlled release, zeta potential, cell viability

## Abstract

In recent years, polymeric materials have been gaining prominence in studies of controlled release systems to obtain improvements in drug administration. These systems present several advantages compared with conventional release systems, such as constant maintenance in the blood concentration of a given drug, greater bioavailability, reduction of adverse effects, and fewer dosages required, thus providing a higher patient compliance to treatment. Given the above, the present work aimed to synthesize polymeric matrices derived from polyethylene glycol (PEG) capable of promoting the controlled release of the drug ketoconazole in order to minimize its adverse effects. PEG 4000 is a widely used polymer due to its excellent properties such as hydrophilicity, biocompatibility, and non-toxic effects. In this work, PEG 4000 and derivatives were incorporated with ketoconazole. The morphology of polymeric films was observed by AFM and showed changes on the film organization after drug incorporation. In SEM, it was possible to notice spheres that formed in some incorporated polymers. The zeta potential of PEG 4000 and its derivatives was determined and suggested that the microparticle surfaces showed a low electrostatic charge. Regarding the controlled release, all the incorporated polymers obtained a controlled release profile at pH 7.3. The release kinetics of ketoconazole in the samples of PEG 4000 and its derivatives followed first order for PEG 4000 HYDR INCORP and Higuchi for the other samples. Cytotoxicity was determined and PEG 4000 and its derivatives were not cytotoxic.

## 1. Introduction

Research on the use of polymeric materials in controlled release systems has progressed in the last two decades through multidisciplinary approaches that seek to improve the therapeutic index and bioavailability of drugs [1]. By definition, the controlled release system (CRS) (or drug delivery system (DDS)) is a delivery system designed to prolong the time of drug release in the body, sustain its plasmatic concentration, and control the temporal and spatial localization of molecules in vivo by applying biological and chemical principles [2]. Drug delivery systems have been designed that deliver the drug to specific sites in the body and depend on the selection of an agent capable of controlling the drug release, sustaining the therapeutic action over time and/or releasing the drug at the level of a particular target tissue or organ. The application of (DDS) is a central strategy to increase the bioavailability, therapeutic efficacy, and safety of the drug [3]. Hence, controlled release systems have advantages when compared with conventional pharmaceutical forms, such as: maintenance of the therapeutic level with low oscillation, preventing toxic levels (as well as local and systemic side effects), avoiding therapeutic sublevels, enabling greater safety in the utilization of some high potency drugs, and providing greater convenience by reducing the number of daily administrations, thus facilitating patient compliance to treatment [4].

Molecular weight polyethylene glycol 4000 (PEG 4000) is a synthetic polymer composed of repeating units of ethylene glycol [3]. It has excellent properties, including biocompatibility [5], solubility in water, simple end-group modification, and non-immunogenic and non-toxic effects. Due to these properties, PEG is used in various fields, especially in pharmaceutical applications [6]. Polyethylene glycol (PEG) acts as a vehicle for other drugs, increasing their half-life. Moreover, the polymer does not possess a drug-like activity; instead, it acts to physically modulate the effects of actual drugs [6]. Ketoconazole (KTZ) is a synthetic imidazole antifungal that has been used for the treatment of superficial and systemic fungal infections [7]. KTZ prevents the synthesis of ergosterol, the major sterol component of fungal plasma membranes, through inhibition of the fungal cytochrome P450-dependent enzyme lanosterol 14-*α*-demethylase. The resulting depletion of ergosterol and concomitant accumulation of 14-a-methylated precursors interferes with the bulk function of ergosterol in fungal membranes and alters both the fluidity of the membrane and the activity of several membrane-bound enzymes (e.g., chitin synthase). The net effect is an inhibition of fungal growth and replication [8].

Scanning electron microscopy (SEM) and atomic force microscopy (AFM) are characterization methods that are applied extremely in polymer science. AFM is used to characterize both the topography and the morphological organization of polymeric materials [9].

The zeta potential (ZP), also termed as the electrokinetic potential, is the potential at the slipping/shear plane of a particle moving under an electric field. The ZP reflects the potential difference between the electric double layer (EDL) of electrophoretic mobile particles and the layer of dispersant around them at the slipping plane. This parameter is used to characterize particle surface charge and to obtain information about their stability [10]. The study of polymers and their pharmacological use aims to reduce or eliminate effects that are undesirable during drug use. In this work, PEG 4000 had its chains modified by organic reactions and its derivatives were incorporated with ketoconazole (Figure 1). The samples were characterized by SEM and AFM to prove the influence of ketoconazole on the morphology of the polymers. The ZP of the PEG 4000 derivatives was measured by electrophoretic mobility to evaluate the steric stability of the samples [11].

## 2. Materials and Methods

The reagents used for the incorporation of ketoconazole were commercial ketoconazole (C_26_H_28_Cl_2_N_4_O_4,_ 531.41 g mol^−1^, meting points 146 °C, pKa 3.0 and 6.5, Log P 3.8, practically insoluble in water, easily soluble in methylene chloride, soluble in methanol, slightly soluble in ethanol) and dichloromethane (CH_2_Cl_2_), purchased from Vetec Química Fina (Duque de Caxias, RJ, Brazil). The poly (vinyl alcohol) and polyethylene glycol (PEG 4000) were purchased from Sigma Aldrich (Saint Louis, MI, USA). The polymers PEG 4000 acetylated, PEG 4000 hydrolyzed, PEG 4000 ethylated, and PEG 4000 halogenated were synthesized by our research group and described elsewhere [9,12,13,14,15].

### 2.1. Incorporation of Ketoconazole in PEG 4000 and Its Derivatives

The synthesis to incorporation was realized in two phases:

Aqueous phase: A beaker containing 40.0 mL of water was warmed on a heating plate. When the temperature reached approximately 70 °C, 0.12 g of poly (vinyl alcohol) was added and slowly stirred until its complete dissolution. Organic phase: On another heater plate, a beaker with 6.00 mL of dichloromethane (CH_2_Cl2) was placed. This solvent solubilized 0.30 g of PEG 4000 (or PEG 4000 derivatives) and 0.10 g of ketoconazole at a temperature of 40 °C. The aqueous phase was poured into the organic phase after the preparation of the two phases. The mixture was subjected to agitation of 500 rpm for 4 h at a temperature of 35 °C. During this process, small portions (1000 mL) of dichloromethane were added. Then, the products were taken to a stove at 40 °C for 24 h to evaporate the dichloromethane. After this time, we obtained solid polymers with varied yields [15]. PEG 4000 derivatives (PEG 4000 acetylate (PEG 4000 ACET), PEG 4000 hydrolyzed (PEG 4000 HYDR), PEG 4000 ethylated (PEG 4000 ETHY), and PEG 4000 halogenated (PEG 4000 HAL)) were chemically synthesized according to the method described previously by our group [10]. The samples with the drug were labeled with an additional “INCORP”.

### 2.2. Characterization Methods

#### 2.2.1. Atomic Force Microscopy (AFM)

Samples for AFM measurements were prepared by the spread-coating method, using 0.6 mg/mL of dichloromethane solution on a mica substrate that was dried after 30 s by a nitrogen flux. The AFM characterization was carried out on a Bruker MultiMode 8 SPM, using the peak force quantitative nanomechanical imaging mode^®^. Si cantilevers (from Bruker), with spring constants of 0.4–0.8 N/m and a tip radius of curvature ~10 nm, were used throughout the study for sample imaging. All AFM images were processed (leveling, profiling, and 3D rendering) using the Gwyddion open-source software [9].

#### 2.2.2. Scanning Electron Microscopy (SEM)

SEM analyses were performed on JEOL equipment JSM6510 and the sample preparation was carried out in a Quorum evaporator. The surface analysis of the samples was performed after they were coated with carbon [8,10].

### 2.3. Determination of Zeta Potential

The electrical charge (zeta potential) of the microparticles in PEG 4000 and its derivatives that were diluted with distilled water was determined by electrophoretic mobility using Zetasizer (Malvern, model Zetasizer Nano series, Nano ZS, Malvern, UK). Each sample was analyzed in triplicate. The results were expressed in millivolts (mV).

### 2.4. Buffer Solution

In the controlled release, one buffer solution of pH 7.3 was used to mimic the blood pH [10].

#### Buffer Solution of pH 7.3

In one beaker, 6.8 g of KH_2_PO_4_ was solubilized in 250 mL of MiliQ water. In another beaker, 1.4 g of NaOH was solubilized in 175 mL of MiliQ water. After both were solubilized, they were mixed, and the volume was made up with MiliQ water to 1000 mL. The solution was then poured into an amber glass container, labeled, and stored in a refrigerator. The pH was measured in a pH meter BEL ENGINEERING [10].

### 2.5. Standard Curve

The standard curve was performed by preparing solutions containing the ketoconazole drug and a pH 7.3 buffer solution at increasing concentrations (10 µg/mL, 20 µg/mL, 30 µg/mL, 40 µg/mL, and 50 µg/mL) in 10 mL volumetric flasks (adapted from [10]). Each solution was analyzed in UV-Vis spectrophotometer GENESYS 10S at the wavelength of 225 nm according to the recommendations by [16].

### 2.6. Controlled Release

Drug delivery systems have been made that deliver the drug to specific sites in the body and then trigger the release of the drug. The incorporation of the drug in the polymeric matrix is important for the release of the drug to occur correctly and is responsible for releasing the drug in specific sites in the body. For controlled release, absorbance readings were taken in the UV spectrophotometer and the wavelength used was 225 nm, since the maximum wavelength of 225 nm was from the drug ketoconazole [16]. The release profile was obtained by soaking duplicate individual samples of PEG 4000 and its derivatives incorporated with ketoconazole. During the release time, the solutions were kept in an oven at 36.5 °C, with stirring. Readings were taken, starting at time 0 (zero) up to 6 h, with aliquots being read every 30 min during the first 4 h and every 1 h thereafter. At each predetermined time interval, the buffer solution was completely drawn and was replaced with a fresh buffer solution adapted from [10]. The summation of each 6 h drug release was considered 100%.

### 2.7. Cell Viability

Macrophages RAW 264.7, cultivated in RPMI 1640 medium (Sigma^®^), were distributed in a 96-well microtiter plate using a density of 5 × 10^5^ cell/well. Then, they were incubated at 37 °C with 5% of CO_2_ for 24 h. The cells were treated with the samples dissolved in RPMI 2% dimethylsulfoxide (DMSO) for 24 h at concentrations ranging from 1000.00 to 7.81 µg/mL. The cell viability was evaluated using the MTT reduction method (3-4,5-dimethyl-thiazol-2-yl-2,5-diphenyltetrazolium bromide). The medium was removed and the wells were washed with RPMI. Then, 100 µL of RPMI without phenol red containing 10% fetal bovine serum and 50 µL of filtered 2 mg/mL MTT were added to the wells. The plates were covered and incubated for 4 h. After this time, the reaction was stopped, using 100 µL of DMSO, and the absorbance of the samples was read in a microplate reader (570 nm). The percentage of cell viability was determined using GraphPad Prism 8.0.1 software.

### 2.8. Statistical Analysis

The results are presented as mean and standard deviation and were analyzed by an analysis of variance (ANOVA), followed by comparison with the Bonferroni test using GraphPad Prism 7.05 software. The significance level was *p* < 0.05.

## 3. Results and Discussion

### 3.1. Atomic Force Microscopy Analyses (AFM) of PEG 4000 and Its Derivatives with and without Incorporation of the Drug

The morphology of polymeric films was observed by AFM and showed changes in the film organization after the drug incorporation. With the AFM technique, it was possible to determine surface morphology, such as roughness parameters, amplitude or height parameters, functional or statistical parameters, and the topography of a given sample [10]. According to Figure 2, it was possible to observe that the polymer structure without incorporation of the drug was characterized by linear chains or linear bundles of chains, which could also be seen in the structures of polymers with the drugs. It could also be seen that samples with the incorporated drug presented higher roughness than those without the drug. This was likely due to the presence of the drug [17].

### 3.2. Scanning Electron Microscopy Analyses (SEM) of PEG 4000 and Its Derivatives with and without Incorporation of the Drug

With SEM, it was possible to notice spheres that formed in some incorporated polymers. According to the photomicrographs, it was possible to notice that ketoconazole had its morphology in the form of crystals. In some micrographs, it was also possible to see several spheres, mainly in the samples PEG 4000 ETHY (Figure 3h) and PEG 4000 HYDR (Figure 3j). Furthermore, it was noted that the PEG 4000 HAL INCORP sample (Figure 3i) did not show microspheres in a representative manner, but the drug had an influence on this polymeric matrix [11].

### 3.3. Determination of the Zeta Potential (ZP)

As expected, the PEG 4000 derivatives exhibited a different surface charge than PEG 4000, but, nevertheless, the values of negative ZP could be attributed to the fatty acid content (a mixture of different esters of behenic acid with glycerol) of the PEG 4000 [18]. Regarding the zeta potential value of the derivatives, when compared with their respective incorporated polymers, it could be seen that there was a significant variation after incorporation of the drug, being statistically different. It is interesting to observe that the samples PEG 4000 ETHY INCORP and PEG 4000 HYDR INCORP showed positive values of ZP as show in Table 1. Ketoconazole is a crystal, therefore these positive charges could be a consequence of it.

### 3.4. Standard Curve

The calibration curve was obtained using ketoconazole in a pH 7.3 buffer solution at concentrations of 10 µg/mL, 20 µg/mL, 30 µg/mL, 40 µg/mL, and 50 µg/mL. The equation of the straight line was obtained by linear regression studies, analyzing the concentrations of ketoconazole and their respective readings. The equation of the line was y = 0.0012x + 0.0378, where x was the concentration in µg/mL and y was the absorbance of the spectrum. Moreover, a linear trend was noted, confirmed by the R^2^ value (0.9963) close to 1 [19].

### 3.5. Controlled Release

For controlled release, the absorbance readings were taken in the UV spectrophotometer; the wavelength used was 225 nm, since the maximum wavelength of 225 nm was from the ketoconazole drug [16]. A 200–400 nm scan was performed to find the maximum wavelength of ketoconazole and then the release was performed. Figure 4.

From the release curves obtained at a wavelength of 225 nm, it was possible to evaluate the behavior of each modified polymer. According to the literature, controlled release systems are characterized by a bimodal or sigmoidal release, which consists of two distinct phases. The first phase is characterized by an accelerated release, so that, in the first hours, there is an up-grade in the release; a phenomenon that occurs in order for the drug to reach the therapeutic concentration range. The second phase consists of a slow release for the maintenance of the drug in the therapeutic window [6]. Thus, all the incorporated polymers used in the controlled release at pH 7.3 represented in Figure 4 presented a continuous release profile, in accordance with the recommended result. Thus, Figure 5 shows a burst in the initial release of ketoconazole from the film. Due to the hydrophilic nature of the polymer used, the polymeric matrix of the gel formed loose channels within the network, causing a fast initial release of drug [20,21,22,23,24].

Based on the cumulative amount of ketoconazole that permeated through the polymeric film, the permeation kinetics were determined. Three kinetic equations were tested: zero order, first order, and Higuchi. Curves were plotted from cumulative permeated ketoconazole versus time (zero order), log cumulative permeated ketoconazole versus time (first order), and cumulative permeated ketoconazole versus the square root of time (Higuchi). The correlation coefficient of each of these permeation kinetics was calculated and compared (Table 2). The permeation kinetic best fit was indicated by the value of the correlation coefficient. As shown in Table 2, the ketoconazole release profiles of the PEG 4000 derivatives followed zero order for the PEG 4000 HYDR INCORP sample and Higuchi for the other samples. The modified release depends on the choice of an agent capable of controlling the release of the drug, sustaining the therapeutic action over time and/or releasing the drug at the level of a particular tissue or target organ, while the polyethylene glycol 4000 (PEG 4000) is employed in the drug release formulations. The lack of a complete release of ketoconazole from the majority of polymer systems studied is directly correlated with the drug–polymer interaction. There has to be an interaction, but it cannot be very strong, since it is possible that the drug will not be released or it will undergo an incomplete release. These models were chosen because they presented the highest R^2^ values. Most of the R^2^ values found were good. Furthermore, according to the work of Elezovic et al., we can consider R^2^ values > 0.8 [24]. In our work, we accepted the model with the highest R^2^ for each sample and, in all of them, the accepted value was higher than 0.8. In the work of Arhewoh et al., the release profile of ibuprofen was studied; they considered values up to 0.7186 in order to describe the drug release profile [25].

These results were consistent with SEM and ZP, which showed that apparently ketoconazole was more superficial in samples PEG 4000 ETHY INCORP and PEG 4000 HYDR INCORP. Therefore, it was released more quickly in these two samples.

### 3.6. Cell Viability

The results obtained in the cytotoxicity test were compared with ISO2009-10993-5 [26], which establishes that a substance is considered cytotoxic when the cell viability is less than 70% [27]. Therefore, the MTT assay provided the data related to polymers according to Figure 6. Based on the results obtained, it was observed that polymers were non-toxic to study cells at concentrations below 250.00 µg/mL, with the exception of acetylated PEG 4000 (non-toxic at concentrations below 125.00 µg/mL), since, in these cases, the cell viability was higher than 70%, therefore considered of low toxicity.

The cytotoxicity of pure ketoconazole was determined to compare with ketoconazole incorporated in the other polymers (Figure 7). It is observed that pure ketoconazole was cytotoxic to cells at most concentrations tested, being non-toxic only at concentrations below 31.25 µg/mL. The cytotoxicity profiles of ketoconazole incorporated into PEG 4000 derivatives varied according to the type of polymer used to carry out the incorporation. Apparently, ketoconazole incorporated in hydrolyzed PEG 4000 showed a profile similar to pure ketoconazole (non-cytotoxic at concentrations lower than 31.25 µg/mL), thus suggesting that this material was not able to reduce the toxicity of the drug. The cytotoxicity of ketoconazole incorporated into acetylated PEG 4000 and halogenated PEG 4000 showed a considerable improvement, since cell viability was greater than 70% at concentrations below 125.00 µg/mL. However, the greatest reduction in drug cytotoxicity was observed when ketoconazole was incorporated into ethylated PEG 4000, since the percentage of cell viability was greater than 70% at concentrations below 250.00 µg/mL, thus suggesting that this would be the most effective polymer matrix indicated to reduce the inherent toxicity of the drug.

## 4. Conclusions

The results of the analyses confirmed the morphological modifications (by AFM and SEM images) of the samples. Based on the data presented, it is possible to state that the incorporation was performed due to the formation of a microsphere. The morphology of polymeric films was observed by AFM and showed changes in the film organization after the drug incorporation. The zeta potential values suggested that the microparticles of PEG 4000 and their derivatives showed a low electrostatic charge, while the cytotoxicity of ketoconazole incorporated into acetylated PEG 4000 and halogenated PEG 4000 showed a considerable improvement, since cell viability was greater than 70%. The release kinetics of ketoconazole in PEG 4000 and derivative samples followed first order for PEG 4000 HYDR INCORP and Higuchi for the other samples. The polymeric matrices of PEG 4000 and its derivatives showed potential for the controlled release of the drug ketoconazole. Furthermore, the formulations were not cytotoxic to human cells.

## Figures and Tables

**Figure 1 polymers-15-02513-f001:**
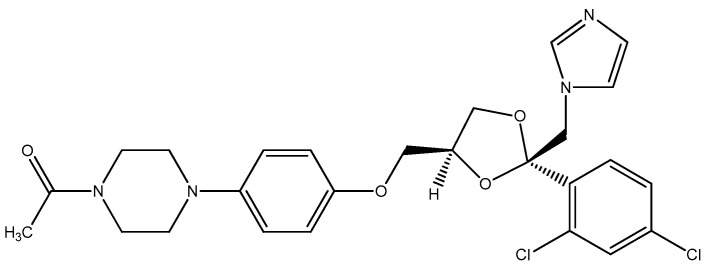
Chemical structure of the ketoconazole drug [11].

**Figure 2 polymers-15-02513-f002:**
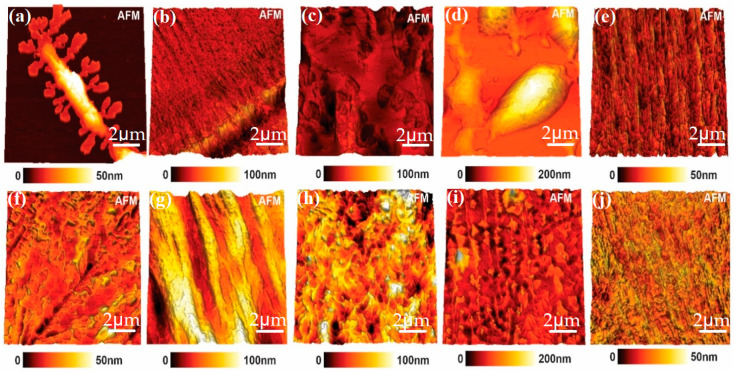
Atomic force microscopy of polymers. PEG 4000 (**a**), PEG 4000 ACET (**b**), PEG 4000 HYDR (**c**), PEG 4000 ETHY (**d**), PEG 4000 HAL (**e**), PEG 4000 INCORP (**f**), PEG 4000 ACET INCORP (**g**), PEG 4000 HYDR INCORP (**h**), PEG 4000 ETHY INCORP (**i**), PEG 4000 HAL INCORP (**j**).

**Figure 3 polymers-15-02513-f003:**
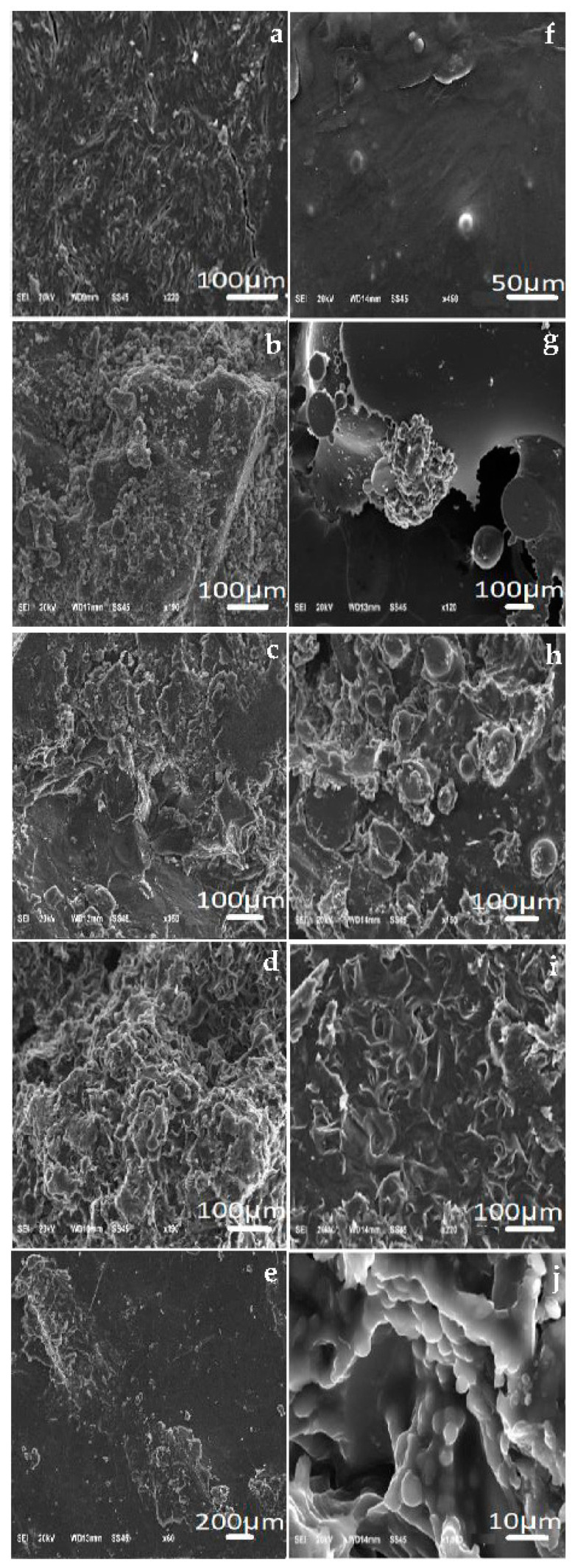
Photomicrographs of the polymers. PEG 4000 (**a**), PEG 4000 ACET (**b**), PEG 4000 ETHY (**c**), PEG 4000 HAL (**d**), PEG 4000 HYDR (**e**), PEG 4000 INCORP (**f**), PEG 4000 ACET INCORP (**g**), PEG 4000 ETHY INCORP (**h**), PEG 4000 HAL INCORP (**i**), PEG 4000 HYDR INCORP (**j**).

**Figure 4 polymers-15-02513-f004:**
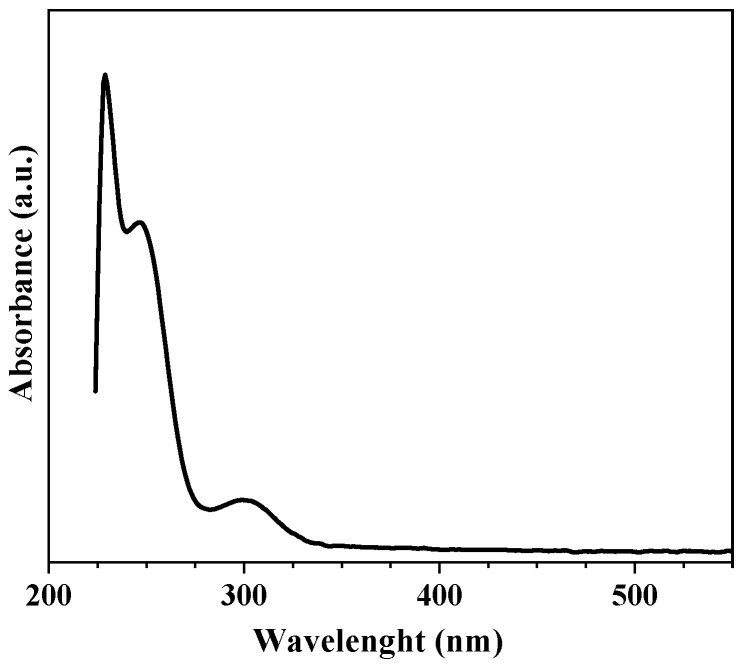
UV/VIS spectra of ketoconazole.

**Figure 5 polymers-15-02513-f005:**
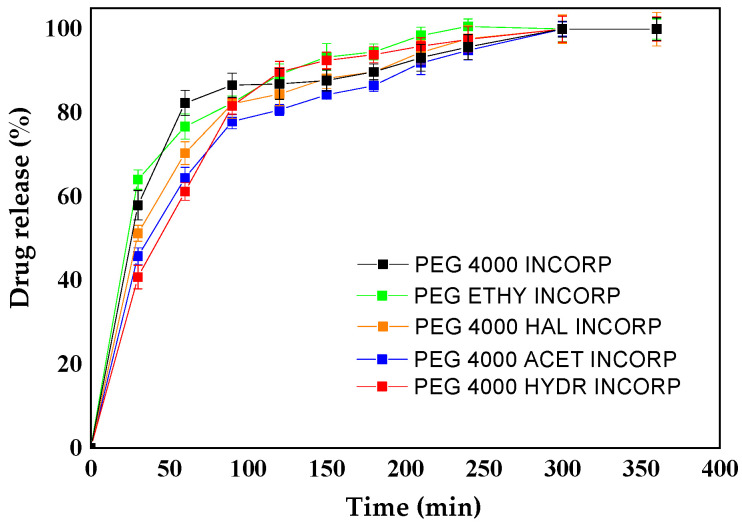
Controlled release graph of the incorporated polymers at pH 7.3 [11].

**Figure 6 polymers-15-02513-f006:**
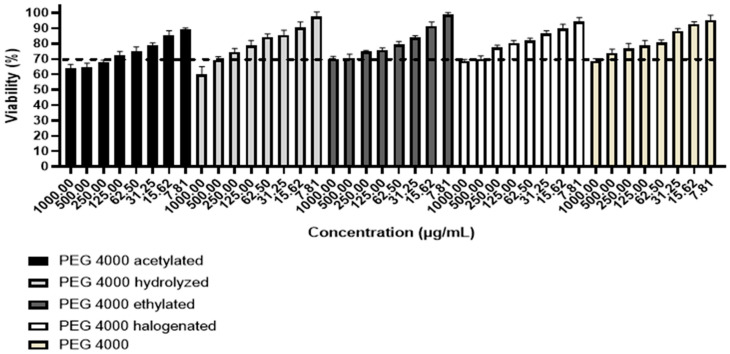
Cell viability of PEG 4000 derivatives.

**Figure 7 polymers-15-02513-f007:**
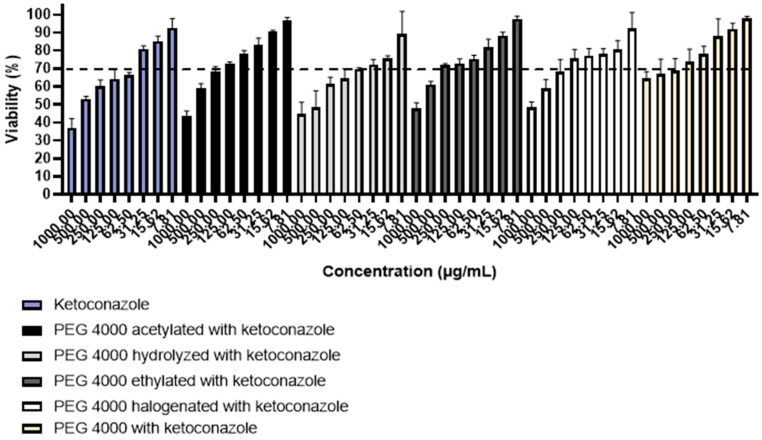
Cell viability of ketoconazole and ketoconazole incorporated in PEG 4000 derivatives.

**Table 1 polymers-15-02513-t001:** Zeta potential of the microparticles of PEG 4000 derivatives.

Sample	Zeta Potential (mV)
PEG 4000	−15.0 ± 2.9 ^a,b^
PEG 4000 ACET	−11.5 ± 0.2 ^a,b^
PEG 4000 HYDR	−6.9 ± 5.5 ^a^
PEG 4000 ETHY	−19.9 ± 3.2 ^b^
PEG 4000 HAL	−9.9 ± 1.5 ^a,b^
PEG 4000 ACET INCORP	−16.2 ± 2.4 ^a,b^
PEG 4000 HYDR INCORP	+10.8 ± 9.0 ^c^
PEG 4000 ETHY INCORP	+4.6 ± 1.3 ^c^
PEG 4000 HAL INCORP	−15.9 ± 4.7 ^a,b^

Values were given as mean ± standard deviations. Mean values with different superscript lowercase letters are significantly different (*p* < 0.05).

**Table 2 polymers-15-02513-t002:** In vitro permeation kinetics of films containing PEG 4000 derivatives.

Sample	Model	*R* ^2^	Equation
PEG 4000 INCORP	zero order	0.8582	Y = 0.0406 X + 3.9932
	first order	0.8421	Y = 0.0067 X + 1.4437
	Higuchi	0.9069	Y = 1.4567 X − 1.7943
PEG 4000 ETI INCORP	zero order	0.9528	Y = 0.0347 X + 6.3991
	first order	0.9401	Y = 0.0042 X + 1.8816
	Higuchi	0.9796	Y = 1.8663 X − 8.2573
PEG 4000 HAL INCORP	zero order	0.9814	Y = 0.0732 X + 5.2562
	first order	0.9619	Y = 0.0079 X + 1.776
	Higuchi	0.9963	Y = 0.9048 X − 1.1621
PEG 4000 ACET INCORP	zero order	0.9911	Y = 0.083 X + 4.7482
	first order	0.9731	Y = 0.0089 X + 1.7202
	Higuchi	0.9997	Y = 0.8038 X − 0.2492
PEG 4000 HYDR INCORP	zero order	0.9999	Y = 0.1361 X + 4.0815
	first order	0.9905	Y = 0.0116 X + 1.7727
	Higuchi	0.9943	Y = 0.4912 X + 1.5557

## Data Availability

Not applicable.

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
