# Peer review of "Controlled Release and Cell Viability of Ketoconazole Incorporated in PEG 4000 Derivatives"

_polymers, 2023, doi:10.3390/polym15112513_

Round 1
Reviewer 1 Report
1. In Fig 1 mentioned reference 11. why?
2. Authors should describr the physico-chemical properties of ketoconazol.
3. In section 2 authors should add purity of ketoconazol.
4. In section 2 authors should add equipment of pH meter and UV-Vis spectrophotometer.
5. Authors should describe why did they use wavelength of 225 nm and show spectra.
6. In Table 2 authors present In vitro permeation kinetics of films containing PEG 4000 derivatives. why coefficient of correclation is low? what is acceptability criteria?
7. Fig 5, 6 have bad quality.
8. Conclusion is not supported by data.
9. What is practical implementation of your work?
5.
Author Response
Response for Reviewer 1
Comments and Suggestions for Authors
1.In Fig 1 mentioned reference 11. why?
Response: reference removed no need.
- Authors should describr the physico-chemical properties of ketoconazol.
Response: The reagents used for incorporation of Ketoconazole (C26H28Cl2N4O4, 531.41 g mol-1, meting points 146 °C, practically insoluble in water, easily soluble in methylene chloride, soluble in methanol, slightly soluble in ethanol) were Commercial...
- In section 2 authors should add purity of ketoconazol.
Response: “...Ketoconazole and dichloromethane (CH2Cl2) purchased from Vetec and the degree of purity of ketoconazole used was 98.31%... “
- In section 2 authors should add equipment of pH meter and UV-Vis spectrophotometer.
Response: pH meter BEL ENGINEERING® and Spectrophotometer genesys 10s uv-vis
2.4.1. Buffer Solution of pH 7.3
In one beaker 6.8 g of KH2PO4 was solubilized in 250 mL of MiliQ water. In another beaker, 1.4 g of NaOH was solubilized in 175 mL of MiliQ water. After both were solubilized, they were mixed and the volume was made up with MiliQ water to 1000 mL. The solution was then poured into an amber glass container, labeled, and stored in a refrigerator. The pH was measured in a pH meter BEL ENGINEERING [10].
2.5. Standard Curve
The standard curve was performed by preparing solutions containing the ketoconazole drug and pH 7.3 buffer solution at increasing concentrations (10 µg/mL, 20 µg/mL, 30 µg/mL, 40 µg/mL, and 50 µg/mL) in 10 mL volumetric flasks (adapted from [10]. Each solution was analyzed in UV-Vis spectrophotometer genesys 10s at the wavelength of 225 nm according to the recommended [18].
- Authors should describe why did they use wavelength of 225 nm and show spectra.
Response: Wavelength of 225 nm is of the ketoconazole
- In Table 2 authors present In vitro permeation kinetics of films containing PEG 4000 derivatives. why coefficient of correclation is low? what is acceptability criteria?
Response: Most of the R2 values found are good. Furthermore, according to the work of Elezovic, Alisa; Elezovic, Amar; and Hadziabdic, Jasmina [The influence of plasticizer in nail lacquer formulations on fluconazole permeability through the bovine hoof membrane. Acta Poloniae Pharmaceutica, 77(1):43-56, 2020], we can consider values > 0.8. In our work, we accepted the model with the highest R2 for each sample, and in all of them, the accepted value was higher than 0.8.
In the work of Arhewoh et al. (Arhewoh, Matthew I; Eraga, Sylvester O; Builders, Philip F; and Uduh, Uchenna A. Snail Mucin-Based Formulation of Ibuprofen for Transdermal Delivery, Journal of Science and Practice of Pharmacy, 1(1):31-36, 2014, which studied the release profile of ibuprofen, they considered values up to 0.7186 to describe the drug release profile.
- Fig 5, 6 have bad quality.
Response: The quality has been increased.
- Conclusion is not supported by data.
Response: Done and changed conclusion
- What is practical implementation of your work?
Response: To synthesize polymeric matrices derived from polyethylene glycol (PEG) capable of promoting the controlled release of the drug ketoconazole, in order to minimize its adverse effects.

Reviewer 2 Report
1. How is your material targeted in special issue?
2. Why is your slow-release experiment done in only one pH medium?
3. Your slow release data has no error analysis, why?
4. Give more refs on the DDS’ descriptions, such as J. Control. Release, 2023, 354, 615–625; J. Mater. Chem. B., 2022, 10, 5105 - 5128; Expert Opin Drug Del., 2022, 19(10), 1183-1202 and New J. Chem., 2022, 46, 13818–13837.
5. Give the TEM and SEM for the full samples.
6. How about the DLS on particle size?
7. Why did you select such carrier, please highlight your design in introduction.
Author Response
Response for Reviewer 2
Comments and Suggestions for Authors
- How is your material targeted in special issue?
Response: Section Polymer Applications and Special Issue Advanced Polymers in Tissue Engineering and Drug Delivery
- Why is your slow-release experiment done in only one pH medium?
Response: Because it simulates blood pH. In the controlled release, one buffer solution of pH 7.3 was used to mimic the blood pH.
- Your slow release data has no error analysis, why?
Response: The authors appreciate for bringing about this important issue. As requested, the error analysis was added to the release graph.
- Give more refs on the DDS’ descriptions, such as J. Control. Release, 2023, 354, 615–625; J. Mater. Chem. B., 2022, 10, 5105 - 5128; Expert Opin Drug Del., 2022, 19(10), 1183-1202 and New J. Chem., 2022, 46, 13818–13837.
Response: Done.
- Xu, Z.; Wu, Z.; Huang S.; Ye, K.; Jian, Y.; Liu , J.; Liu, J.;, Xinwu Lu, X.; Li, B. All rights reserved.A metal-organic framework-based immunomodulatory nanoplatform for anti-atherosclerosis treatment. Control. Release 2023, 354, 615–625.
- Li, M.; Yin,S.; Lin,M.; Chen, X.; Pan, ; Peng, Y.; Sun, J.; Kumar, A.; Liu, J.;Current status and prospects of metal–organic frameworks for bone therapy and bone repair.J. Mater. Chem. B.,2022,10, 5105 – 5128.
- Rao, C.; Liao, D.; Pan, Y.; Zhong, Y.; Zhang, W.; Ouyang, Q. Novel formulations of metal-organic frameworks for controlled drug delivery. Expert Opin Drug Del. 2022, 19(10), 1183-1202.
- Liu, S..; Qiu,Y.; Liu, Y.; Zhang, ; Dai, Z.; Srivastava, D.; Kumar, A.; Ying Pan, Y.; Liu, J. Recent advances in bimetallic metal–organic frameworks (BMOFs): synthesis, applications and challenges.New J. Chem. 2022, 46, 13818–13837.
- Give the TEM and SEM for the full samples.
Response: Scanning Electron Microscopy Analyses (SEM) of PEG 4000 and its derivatives with and without incorporation of drug
- How about the DLS on particle size?
Response: No particle size analysis was performed by DLS. Size analysis was not performed as the particles precipitate rapidly in water, therefore, size analysis by DLS would not provide reliable results.
- Why did you select such carrier, please highlight your design in introduction.
Response: Molecular weight polyethylene glycol 4000 (PEG 4000) is a synthetic polymer composed of repeating units of ethylene glycol. It has excellent properties, including biocompatibility, solubility in water, simple end-group modification, non-immunogenic, and non-toxic. Due to these properties, PEG is used in various fields, especially in pharmaceutical applications.

Round 2
Reviewer 1 Report
I have a serious consideration in publishing that paper in Polymers due to several reasons.
1. Authors should add pKa, Log P of ketoconazol.
2. Authors did not describe correctly why did they use wavelength of 225 nm and did not show spectra. If no spectra, so we may think that all research could be falsified.
3. Authors did not explain correctly why coefficient of correlation is low and what is acceptability criteria?
In general, presented research is very simple and i not novel.
Reviewer 2 Report
accept
Author Response
Response for Editor, Reviewer 1 and Reviewer 2-Round 2
We appreciate the comments and suggestions made about our manuscript. We have revised the manuscript. We believe these suggestions substantially improved this new version of the manuscript. All changes are highlighted in red, to ease their prompt identification. Some of our more detailed responses to the reviewers are commented here below in this letterThank you very much for all your kind attention.
Sincerely,
Dra. Viviane Martins Rebello dos Santos
Comments and Suggestions for Editor, Reviewer 1 and Reviewer 2
- We notice that after a round of revisions, the main text part of the paper (3432 words) is still not quite up to the suggested minimum word count of 4000 words (https://www.mdpi.com/about/article_types). In the second round of revision, please try to extend the content by considering these points: Referring to reviewers' comments to add content to the text, adding fully experimental details, presenting completely all the results, and providing comprehensively the background and overview of the research in the introduction section.
Response: Done.
- Please revise the manuscript according to the referees' comments and
upload the revised file within 7 days.
Response: Done.
- Please use the version of your manuscript found at the above link for your
revisions.
(I) Please check that all references are relevant to the contents of the
manuscript.
Response: Done.
(II) Any revisions to the manuscript should be marked up using the “Track
Changes” function if you are using MS Word/LaTeX, such that any changes can
be easily viewed by the editors and reviewers.
(III) Please provide a cover letter to explain, point by point, the details
of the revisions to the manuscript and your responses to the referees’
comments.
(IV) If you found it impossible to address certain comments in the review
reports, please include an explanation in your appeal.
(V) The revised version will be sent to the editors and reviewers.
reports, please include an explanation in your appeal.
Response: Done.
Comments and Suggestions for Reviewer 1
I have a serious consideration in publishing that paper in Polymers due to several reasons.
1.Authors should add pKa, Log P of ketoconazol.
Response: Added “Ketoconazole is a dibasic weak base with pKa = 6.5 and 3.0. and LogP 3.8.”
- Authors did not describe correctly why did they use wavelength of 225 nm and did not show spectra. If no spectra, so we may think that all research could be falsified.
Response: For controlled release, absorbance readings were taken in the UV spectrophotometer and the wavelength used was 225 nm, because is the Maximum Wavelength of 225 nm is of Ketoconazole drug[18]. A 200-400 nm scan was performed to find the maximum wavelength of ketoconazole and then make release, Figure 4
Figure 4. UV/VIS spectra of ketoconazole.
- Authors did not explain correctly why coefficient of correlation is low and what is acceptability criteria?
Response: Modified release depend on the choice of an agent capable of controlling the release of the drug, sustaining the therapeutic action over time, and/or releasing the drug at the level of a particular tissue or target organ and Polyethylene glycol 4000 (PEG 4000) is employed in drug release formulations. The lack of complete release of ketoconazole from the majority of polymer systems studied is directly correlated with drug–polymer interaction. There has to be an interaction, but it cannot be very strong as it is possible that the drug will not be released or undergo an incomplete release. These models were chosen because they presented the highest R2 values. Most of the R2 values found are good. Furthermore, according to the work of Elezovic et al., we can consider R2 values > 0.8 [27]. In our work, we accepted the model with the highest R2 for each sample, and in all of them, the accepted value was higher than 0.8. In the work of Arhewoh et al., studied the release profile of ibuprofen, they considered values up to 0.7186 to describe the drug release profile[28].
- Elezovic, A.; Elezovic, A.; Hadziabdic, J. The influence of plasticizer in nail lacquer formulations on fluconazole permeability through the bovine hoof membrane. Acta Poloniae Pharmaceutica 2020, 77(1):43-56.
- Arhewoh, M. I.; Eraga, Sylvester, O.; Builders, P. F.; Uduh, U. A. Snail Mucin-Based Formulation of Ibuprofen for Transdermal Delivery, Journal of Science and Practice of Pharmacy 2014, 1(1):31-36.
Round 3
Reviewer 1 Report
Authors should include UV/VIS spectrum of ketoconazole in main text
Author Response
Dear Editor
We appreciate the comments and suggestions made about our manuscript. We have revised the manuscript and include in main text. We believe these suggestions substantially improved this new version of the manuscript. All changes are highlighted in red, to ease their prompt identification. Some of our more detailed responses to the reviewers are commented here below in this letter.
Thank you very much for all your kind attention.
Sincerely,
Response for Reviewer 1
Authors: should include UV/VIS spectrum of ketoconazole in main text
Response: Done in main text.
Dra. Viviane Martins Rebello dos Santos
